# Best of mini-N in-loop Sampling: A Contextual Quality Reward Model for Reliable and Efficient Best-of-N Sampling

## Abstract

Modern preference alignment techniques, such as Best-of-N (BoN) sampling, rely on reward models trained with pairwise comparison data. While effective at learning relative preferences, this paradigm fails to capture a signal of response acceptability, leaving systems vulnerable to selecting the least bad of many unacceptable options. This is particularly problematic for hard prompts, where the risk of such false acceptances increases with the number of samples. In this paper, we address this critical reliability gap by introducing a new data collection and modeling framework. By augmenting preference data with an outside option, inspired by discrete choice models, we train a reward model that can distinguish not just what is *better*, but what is *good enough*. We leverage this capability to create an adaptive inference strategy, best of mini-N in-loop, which partitions the generation budget into sequential loops with a calibrated, early-exit condition. Our experiments show that when tuned as an alignment guardrail, it reduces reliability failures by 70%, and when tuned as an inference accelerator, it improves average inference speed by over 22% in IMDB-sentiment setting. We thus provide a principled and flexible framework for practitioners to explicitly manage the trade-off between reliability and computational efficiency.

## 1 Introduction

The capabilities of Large Language Models (LLMs) have been significantly enhanced through alignment with human feedback, making them more helpful and harmless (Bai et al., 2022), instruction-following (Ouyang et al., 2022), and capable of sophisticated tasks (Ziegler et al., 2019). Such advances are driven by preference-based alignment methods, ranging from inference-time techniques like Best-of-N (BoN) sampling (Nakano et al., 2021; Ouyang et al., 2022) to post-training optimization approaches like Direct Preference Optimization (DPO) (Rafailov et al., 2023). Crucially, the effectiveness of these approaches depends entirely on the data used to train their underlying preference models.

Current methods predominantly rely on pairwise comparison datasets, where humans select the better of two responses. This paradigm, rooted in models like Bradley–Terry (Bradley & Terry, 1952), has an inherent limitation: it only captures relative preference, not response acceptability. While the field often implicitly treats the chosen response as "desirable" (Jung et al., 2024; Ethayarajh et al., 2024), the data merely indicates that it is better than the alternative. As a result, a reward model trained this way can rank responses but cannot determine whether a response is acceptable or unacceptable (Wang et al., 2023). This limitation critically undermines BoN sampling. There is no guarantee that the best of $N$ options meets a minimum quality threshold. It is possible that BoN simply selects the least bad of many poor options.

In this paper, we address this gap by introducing a new data collection protocol that extends standard pairwise comparisons with an **outside option**. By allowing annotators to reject all candidate responses, our method captures a direct signal of contextual acceptability. We show that a reward model trained on this richer data can distinguish between responses that are merely *better* and those that are genuinely *acceptable* within the context. This seemingly simple modification unlocks several critical capabilities that we demonstrate in this paper:

First, we present a theoretical analysis and empirical evidence showing that standard Best-of-N (BoN) sampling is increasingly unreliable for challenging prompts. As $N$ grows, the probability of a false acceptance (i.e., selecting the least-bad candidate that still fails a minimum quality threshold) rises substantially. Our experiment results show that the number of such reliability failures in the BoN baseline more than doubles as $N$ increase from 1 to 32.

Building on this, we introduce a novel adaptive inference strategy, **Best of mini-N in-loop,** which leverages the signal of contextual acceptability captured by our reward model. This single, flexible framework partitions a large generation budget into smaller, sequential loops and checks if an acceptable response has been found, enabling an early exit. This strategy is powerful because it can be tuned for two distinct goals: serving as a robust guardrail for reliability-critical tasks or as an efficiency optimizer for speed-critical ones.

For reliability critical applications, the framework can be configured as what we term an **Alignment Guardrail.** By setting a calibrated quality threshold, the system refrains from outputting a response unless it is demonstrably acceptable. This is essential in contexts like customer-facing chatbots. For instance, rather than providing a policy-compliant but unhelpful response (the least bad option it could generate), the system can recognize that no candidate will satisfy the user and instead escalate the query to a human agent. This prevents user frustration and ensures a higher standard of interaction. Our experiments confirm this is highly effective, reducing the number of false acceptances by **70%** compared to the standard BoN baseline. This ability to abstain or invoke fallback strategies is a crucial capability for safe and reliable systems (Lightman et al., 2024; Snell et al., 2024; Kang et al., 2025; Bai et al., 2022).

Conversely, for speed critical applications where a slight misalignment is tolerable, such as document summarization, the same framework can be configured as a fast **Inference Accelerator**. Here, the goal shifts from finding the best response to finding the first acceptable one. By terminating the generation process as soon as a good-enough candidate is identified, our method achieves the fastest possible inference time. Our experiments show this approach outperforms the baseline by over **22%** on average.

This establishes a clear and tunable trade-off between reliability and computational efficiency that is absent in prior methods.

## 2 Related Work

Modern LLM alignment techniques, most prominently Reinforcement Learning from Human Feedback (RLHF) and RL-free direct methods like Direct Preference Optimization (DPO), are trained on datasets of pairwise preferences (Ouyang et al., 2022; Rafailov et al., 2023). Such data is inherently ordinal, so reward models can rank candidates, but lack an absolute acceptability notion, which allows "least-bad" selections on hard prompts. The notion of incorporating an outside option into preference data has been previously noted by Wang et al. (2024). In particular, the *HelpSteer2-Preference* dataset introduces a "Neither response is valid" label, functioning as an implicit outside option when both candidate responses are unacceptable. However, this signal was employed solely as a data-cleaning heuristic, samples with this label were excluded during preprocessing rather than integrated into the reward modeling process.

Our work departs from prior approaches by treating the outside option not as a data cleaning heuristic, but as a core learning signal. We use this signal within a discrete choice framework to explicitly teach the reward model the concept of a contextual "acceptability threshold." This allows the model to distinguish not just what is *better*, but what is *good enough*, a critical capability for reliable inference-time control.

BoN is an effective but compute-intensive approach to improving inference-time quality. Inference-time selection methods, including BoN and reward-model reranking, raise output quality without modifying the base policy, as demonstrated by rejection sampling against a reward model in *WebGPT* (Nakano et al., 2021) and by broader alignment experiments on preference-modeling and scaling (Askell et al., 2021). Nevertheless, when all samples are weak, these methods can still return the least-bad candidate. To reduce computational burden, recent work prunes low-promise generations mid-decode via speculative rejection sampling (Sun et al., 2024), adapts the sample size $N$ to prompt difficulty in Adaptive BoN (Raman et al., 2025), introduces confidence-based early exit for generation (Schuster et al., 2022), and accelerates decoding through speculative decoding while preserving the output distribution (Leviathan et al., 2023).

Our best-of-mini-$N$ in-loop strategy is complementary: instead of pruning partial candidates or fixing $N$ in advance, it sequences small batches and terminates as soon as a candidate exceeds a context-calibrated acceptability threshold, yielding a principled, quality-based stopping rule. Although we leave integration to future work, the procedure is naturally compatible with pruning, adaptive $N$, early-exit, and speculative decoding, and can be combined with these techniques for additional efficiency gains.

## 3   A Choice-Based Reward Model

We develop a choice-based reward model by adapting McFadden's discrete-choice model (McFadden, 2001) to include an explicit "outside option" that captures a labeler's decision to reject all candidates. We then specify the utility model and derive the resulting multinomial logit choice probabilities. From these probabilities, we construct a normalized reward anchored to the outside option and analyze its properties: the reward preserves relative preference rankings while supplying an interpretable, context-dependent acceptability signal. Under standard identification conditions, this normalized reward is uniquely recoverable from choice data.

### 3.1   Model Specification

Consider a scenario where a human labeller is presented with a choice set $\mathcal{Y} = \{y_0, y_1, ..., y_J\}$. The options $y_i$ for all $i \neq 0$ are responses generated by a policy $\pi(\cdot|x)$ for a given prompt $x$. Option $y_0$ is a symbolic representation of the outside option, signifying the choice to reject all $y_i$ for $i \neq 0$.

We model the utility a labeller derives from each option $y_i$ as the sum of a deterministic reward component, $\tilde{R}(x, y_i)$, and a stochastic error term, $\epsilon_i$:

$$U(x, y_i) = \tilde{R}(x, y_i) + \epsilon_i.$$

Following standard discrete choice literature, we assume the error terms $\epsilon_i$ are independent and identically distributed according to a Gumbel distribution with zero location and unit scale. A rational labeller selects the option with the highest utility. That is, option $y_i$ is chosen if:

$$\tilde{R}(x, y_i) + \epsilon_i \geq \max_{j \leq J} \left( \tilde{R}(x, y_j) + \epsilon_j \right). \tag{1}$$

Under this assumption, the probability of the labeller choosing option $y_i$, denoted by the indicator $d_i = 1$, takes the familiar multinomial logit form:

$$
\begin{aligned}
\Pr(d_i = 1 | x, \mathcal{Y}) &= \frac{\exp(\tilde{R}(x, y_i))}{\sum_{j=0}^{J} \exp(\tilde{R}(x, y_j))} \\
&= \frac{\exp(\tilde{R}(x, y_i) - \tilde{R}(x, y_0))}{\sum_{j=0}^{J} \exp(\tilde{R}(x, y_j) - \tilde{R}(x, y_0))} \\
&= \frac{\exp(R(x, y_i))}{1 + \sum_{j=1}^{J} \exp(R(x, y_j))}
\end{aligned} \tag{2}
$$

where we define the normalized reward as

$$R(x, y_i) \equiv \tilde{R}(x, y_i) - \tilde{R}(x, y_0).$$

Consequently, the normalized reward for the outside option is $R(x, y_0) = 0$. Because the errors are i.i.d. Gumbel, the induced model satisfies the independence of irrelevant alternatives; in our fixed-prompt setting with an explicit outside option this is a tolerable approximation, but tasks with correlated alternatives would call for a richer error structure.

The utility of the outside option, $\tilde{R}(x, y_0)$, is not tied to a specific generated response. Instead, we model it as a prompt-dependent rejection threshold, $C(x)$. This is a critical assumption: A labeller's tolerance for imperfect responses—and thus their threshold for what is "good enough"—naturally varies with the

prompt's context. For example, a fact-based question demanding precision will have a much higher rejection threshold than a creative prompt where minor errors are tolerable. Our normalized reward, $R(x, y_i)$, therefore represents the quality of a response measured against this prompt-specific, contextual standard of acceptability. We do not estimate $C(x)$ separately; we learn $R(x, y)$ directly and use $R(x, y) > 0$ as the acceptability criterion.

## 3.2 Properties and Advantages

### 3.2.1 Invariance of Relative Preference

The normalization of the reward function does not alter its primary function for preference ranking. The subtraction of the outside option's utility, $\tilde{R}(x, y_0) = C(x)$, is an affine transformation applied to all potential responses for a given prompt $x$. This ensures that the relative preference ordering between any two generated responses, $y_i$ and $y_j$, is preserved. That is, $\tilde{R}(x, y_i) > \tilde{R}(x, y_j)$ if and only if $R(x, y_i) > R(x, y_j)$. Therefore, the model's capacity to function as a preference-based reward signal for standard alignment techniques remains intact.

### 3.2.2 Identification of Contextual Acceptability

A significant advantage of incorporating an outside option is that the normalized reward model $R(x, y_i)$ is fully identified. Unlike standard pairwise comparison models such as the Bradley-Terry model, whose reward functions are only identified up to an arbitrary constant $c$ (e.g., $R(x, y_i) + c$) (Wang et al., 2023), our model is anchored by the outside option.

This can be seen by rearranging Eq. equation 2 to express the normalized reward directly in terms of choice probabilities:

$$R(x, y_i) = \log(\Pr(d_i = 1 | x, \mathcal{Y})) - \log(\Pr(d_0 = 1 | x, \mathcal{Y}))$$

The reward $R(x, y_i)$ is the log-odds of choosing response $y_i$ over the outside option, a quantity that can be directly estimated from choice data. This complete identification makes a crucial difference in interpretation. A Bradley-Terry model can only reveal which response is better (relative preference), whereas our model reveals whether a response is good in a contextual sense. The criterion for this is the sign of the normalized reward:

- If $R(x, y_i) > 0$, the utility of response $y_i$ exceeds the rejection threshold $C(x)$ and can be considered acceptable.

- If $R(x, y_i) < 0$, the utility is below the threshold and the response can be considered unacceptable.

This provides a clear, interpretable measure of response quality that goes beyond simple pairwise ranking.

## 4 Estimation & Evaluation

This section outlines the empirical framework for estimating the parameters $\theta$ of our reward model and evaluating its performance. We begin by defining the log-likelihood function used for training, which is derived directly from the choice probabilities established in Section 3. Next, to assess the model's core capability of identifying context acceptability, we introduce a novel evaluation protocol. We detail how the preference data can be transformed for a binary classification task and discuss key metrics—such as Precision and False Positive Rate (FPR) for its role as a robust alignment guardrail, and Recall for its effectiveness as a fast inference Accelerator. Finally, we address the important issue of class imbalance, arguing that the observed choice proportions are a vital diagnostic signal of the generator's quality and should not be artificially altered through re-sampling.

### 4.1 Log-Likelihood Function

We estimate the parameters $\theta$ of the reward function $R_\theta(x, y)$ using Maximum Likelihood Estimation (MLE). The objective is to find the parameters that maximize the probability of observing the choices made by human labellers in our dataset, $\mathcal{D}$. The dataset consists of $N$ observations, where each observation $k$ is a tuple $(x_k, \mathcal{Y}_k, d_k)$. Here, $x_k$ is the prompt, $\mathcal{Y}_k = \{y_{k,0}, y_{k,1}, \ldots, y_{k,J_k}\}$ is the choice set presented to the labeller, and $d_k \in \{0, 1, \ldots, J_k\}$ is the index of the chosen option. The size of the choice set, $J_k$, can vary for each observation, with $J_k \geq 1$.

The total log-likelihood of the dataset is the sum of the log-probabilities for each observed choice:

$$\mathcal{L}(\theta; \mathcal{D}) = \sum_{k=1}^{N} \log \Pr(d_k | x_k, \mathcal{Y}_k, \theta) \tag{3}$$

By substituting the choice probability from Eq. equation 2, the log-likelihood for a single observation where the labeller chose option $d_k$ is given by:

$$\log \Pr(d_k | x_k, \mathcal{Y}_k, \theta) = R_\theta(x_k, y_{k,d_k}) - \log \left( 1 + \sum_{j=1}^{J_k} \exp(R_\theta(x_k, y_{k,j})) \right)$$

Note that the '1' in the denominator of the log term corresponds to $\exp(R_\theta(x_k, y_{k,0}))$, as the normalized reward of the outside option is zero by definition. The model is trained by maximizing the total log-likelihood in Eq. equation 3 using a standard gradient-based optimizer.

### 4.2 Evaluation Metrics

To evaluate the model's ability to identify response acceptability within the context, we reframe the evaluation as a binary classification task. This allows us to use standard classification metrics to assess how well the learned reward function, $\hat{R}_\theta(x, y)$, distinguishes between acceptable and unacceptable responses.

#### 4.2.1 Data Transformation for Binary Evaluation

First, we transform our original evaluation dataset, $\mathcal{D}_e$, into a binary classification dataset, $\mathcal{D}_{bin}$. Each observation in the original dataset is processed as follows:

1. **If a response was chosen ($d_k > 0$):** The labeller's choice reveals the preference $U(x_k, y_{k,d_k}) \geq U(x_k, y_{k,0})$. This corresponds to a single binary event where the chosen response is preferred over the outside option. We therefore map this observation to a single positive instance in the new dataset: $(x_k, y_{k,d_k}, 1)$, where 1 is the ground truth label for "acceptable."

2. **If the outside option was chosen ($d_k = 0$):** This choice reveals that the utility of the outside option was greater than that of every generated response: $U(x_k, y_{k,0}) \geq U(x_k, y_{k,j})$ for all $j \in \{1, \ldots, J_k\}$. This single observation unpacks into $J_k$ distinct pairwise preferences. We therefore map this observation to $J_k$ negative instances: $\{(x_k, y_{k,j}, 0)\}_{j=1}^{J_k}$, where 0 is the ground truth label for "unacceptable."

It is important to note that this data transformation uses the observed human choice as a proxy for the true, latent quality of a response. Ideally, the ground truth label for a response $y$ would be determined by the sign of its true reward, i.e., $R(x, y) > 0$, as this represents the average preference across a population of labellers, not just the stochastic utility of a single individual. However, due to the stochastic nature of the choice model specification equation 1, we only observe a noisy realization of this preference. The metrics are therefore conditioned on the observed choice, $d_k$, rather than the unobserved reward sign. For a given threshold $\tau$ (which is 0 in our framework), we are essentially using the labeller's choice to approximate the ideal but unobservable target. Instead of directly measuring a quantity like $\Pr(\hat{R}_\theta(x, y) > \tau | R(x, y) > 0)$, our metrics are based on the observable proxy $\Pr(\hat{R}_\theta(x, y) > \tau | \text{response } y \text{ was chosen})$. This is the best available approximation of the desired target.

### 4.2.2 Probabilistic Model and Metrics

Given this binary dataset and the choice model specification equation 1, we can assess the model's performance. The probability of a response $y$ being acceptable is modeled as $P(1|x, y) = \sigma(\hat{R}_\theta(x, y))$, where $\sigma(\cdot)$ is the sigmoid function. The probability of it being unacceptable is $P(0|x, y) = 1 - \sigma(\hat{R}_\theta(x, y)) = \sigma(-\hat{R}_\theta(x, y))$.

From this, we can compute several key metrics to understand different aspects of the model's performance:

- **Precision:** Measures the proportion of responses identified as acceptable ($R_\theta > 0$) that are truly acceptable (label 1). High precision is the key statistic for the alignment guardrail, ensuring the model is trustworthy when it approves a response. This minimizes the risk of falsely endorsing a candidate that falls short of the labeller's standard of acceptability.

- **Recall:** Measures the model's capacity to identify all truly acceptable responses. This metric is the primary driver of the inference accelerator's performance. A model with high recall excels at finding an acceptable candidate swiftly, maximizing the probability of an early exit and thereby slashing unnecessary computational overhead.

- **False Positive Rate (FPR):** Measures the proportion of unacceptable responses (ground truth label 0) that are incorrectly classified as acceptable ($\hat{R}_\theta > 0$). A low FPR is the most critical measure of the alignment guardrail's strictness. It directly quantifies the escape rate-the frequency at which misaligned responses are shown to the user—making it fundamental to ensuring reliability and trust.

These metrics serve as key examples, and the framework allows researchers to employ any standard binary classification metric to evaluate the model's performance beyond simple preference ranking.

### 4.3 A Note on Class Imbalance and Choice-Based Sampling

This subsection addresses the potential issue of class imbalance when using an outside option and argues that researchers should not attempt to correct for it through re-sampling.

In standard preference models without an outside option, such as the Bradley-Terry model, class imbalance is not a primary concern. The responses $\{y_1, \ldots, y_J\}$ for a prompt are typically drawn i.i.d. from a generator policy $\pi(\cdot|x)$. As the index of a response carries no intrinsic meaning, the population proportion of each response being chosen, $Q(i)$, should be equal for all $i \in \{1, \ldots, J\}$. Ideally, $Q(i) = 1/J$, meaning no inherent imbalance exists to be corrected.

However, the introduction of an outside option fundamentally changes this dynamic. The proportion of the outside option being chosen, $Q(0)$, can be substantially different from the choice proportions of the generated responses, $Q(i)$ for $i > 0$. While it may be tempting to rebalance the dataset to equalize these proportions, this should be avoided.

Attempting to correct for this imbalance via re-sampling techniques introduces a form of sampling bias. As demonstrated by Manski & Lerman (1977), estimating a choice model on a dataset where the choice proportions have been artificially altered leads to inconsistent estimates of the underlying reward function. Specifically, if one uses an altered sampling distribution $H(i)$ instead of the true population distribution $Q(i)$ to estimate the model with the maximum likelihood objective in Eq. equation 3, the resulting reward function, $R'$, becomes a biased estimator of the true reward $R$:

$$R'(x, y_i) = R(x, y_i) + \left[ \log\left(\frac{H(i)}{Q(i)}\right) - \log\left(\frac{H(0)}{Q(0)}\right) \right].$$

The bracketed term represents a bias that is a function of the artificial sampling strategy. This bias corrupts the reward's scale and offset, invalidating its interpretation as a measure of contextual acceptability. Therefore, manual rebalancing of the dataset must be avoided.

### 4.3.1 Interpreting Observed Imbalances

Instead of being a problem to be corrected, a significant class imbalance between the outside option and the generated responses should be treated as a valuable diagnostic signal about the quality of the generator policy $\pi(\cdot|x)$:

- **If $Q(0)$ is very high:** This indicates that the base policy is frequently incapable of producing responses that surpass the labellers' acceptance threshold. The correct remedy is not to down-sample the outside option, but to improve the generator policy itself, for instance, through another round of Supervised Fine-Tuning (SFT) on high-quality data.

- **If $Q(0)$ is very low:** This suggests that the base policy is already strong and almost always produces acceptable responses. In such cases, the signal of contextual acceptability provided by the outside option is less informative, as nearly everything is good. A simpler, standard Bradley-Terry preference model may be sufficient for learning the relative ranking between these high-quality responses.

The primary takeaway is that the observed choice proportions are a crucial source of information about the generator policy's quality and should not be obscured through artificial rebalancing.

## 5 Best of mini-N in-loop

This section introduces our primary contribution: an adaptive inference strategy we term best of mini-N in-loop. We show how this single framework can be configured to serve two distinct purposes: as a robust alignment guardrail to maximize reliability, or as a fast inference accelerator to maximize speed. While both configurations are critical, this section's primary focus is on developing the theory and mechanics behind the Alignment Guardrail.

We begin by providing a mathematical analysis of a critical failure mode in standard BoN, demonstrating how the probability of a false acceptance inflates for difficult prompts as more candidates are sampled. We then present our in-loop method and its core mechanism for the guardrail: a sequence of calibrated, adaptive thresholds designed to control this risk. Finally, we discuss how this approach allows practitioners to navigate the explicit trade-off between the reliability offered by the guardrail and the latency improvements of the accelerator.

### 5.1 The Failure Mode of Standard BoN

The purpose of this subsection is to demonstrate that for difficult prompts, the standard BoN sampling process is increasingly prone to false acceptances when $N$ is not sufficiently large enough. We define a prompt $x$ as hard if the generator policy $\pi(\cdot|x)$ is unlikely to produce an acceptable response, meaning the probability of any single response being good, $p_g = \Pr_{y \sim \pi(\cdot|x)}(R(x,y) > 0)$, is very small. For our analysis, we consider a single, representative hard prompt.

The BoN process draws $N$ i.i.d. samples $\{y_1, \ldots, y_N\}$ from $\pi(\cdot|x)$ and selects the one with the highest estimated reward, which we denote as $y^*$:

$$y^* = \arg \max_{i \in \{1, \ldots, N\}} \hat{R}_\theta(x, y_i).$$

We are interested in the probability of a false acceptance, $P_{FA}(N)$, which is the joint probability that the winning response is incorrectly deemed acceptable while being truly unacceptable:

$$P_{FA}(N) = \Pr(\hat{R}_\theta(x, y^*) > 0 \text{ and } R(x, y^*) < 0 | N).$$

To analyze how this probability changes with $N$, we can decompose it using the law of total probability by conditioning on the number of good responses, $K$, in the set of $N$ samples. Let $K = k$, where $k \in \{0, \ldots, N\}$.

The probability of this event, $q_N(k) = \Pr(K = k|N)$, follows a binomial distribution: $q_N(k) = \binom{N}{k}p_g^k(1 - p_g)^{N-k}$. The total probability of false acceptance is then:

$$P_{FA}(N) = \sum_{k=0}^{N} \Pr(\text{false acceptance}|N, K = k) \cdot q_N(k).$$

Let's denote the conditional probability as $a_{k,N} \equiv \Pr(\text{false acceptance}|N, K = k)$. This term represents the probability of a false acceptance given a pool of $k$ good and $N - k$ bad candidates. The central insight is that for a fixed number of good candidates $k$, increasing the total number of samples $N$ can only increase this probability. Adding another bad candidate to the pool can either leave a previous bad winner in place or replace it with a new bad winner that scored even higher due to estimation noise. It cannot, however, cause a previously false-positive outcome to resolve correctly. Thus, $a_{k,N}$ is monotonically non-decreasing in $N$ for any fixed $k$:

$$a_{k,N+1} \geq a_{k,N} \quad \text{for } k \leq N. \tag{4}$$

To formally analyze the behavior of $P_{FA}(N)$, we examine the difference $P_{FA}(N + 1) - P_{FA}(N)$. Using the binomial recurrence relation $q_{N+1}(k) = (1 - p_g)q_N(k) + p_g q_N(k - 1)$, where $p_b \equiv 1 - p_g$, we can write:

$$\begin{aligned}
P_{FA}(N + 1) &= \sum_{k=0}^{N+1} a_{k,N+1}q_{N+1}(k) \\
&= \sum_{k=0}^{N+1} a_{k,N+1}[p_b q_N(k) + p_g q_N(k - 1)] \quad (\text{with } q_N(-1) = q_N(N + 1) = 0) \\
&= a_{N+1,N+1}p_g q_N(N) + \sum_{k=0}^{N} a_{k,N+1}[p_b q_N(k) + p_g q_N(k - 1)] \\
&= \sum_{k=0}^{N} a_{k,N+1}[p_b q_N(k) + p_g q_N(k - 1)].
\end{aligned}$$

Note that $a_{N+1,N+1}p_g q_N(N)$ is zero by definition. A false acceptance requires selecting an unacceptable response, which is impossible when the pool of candidates contains only acceptable options. The difference is then found by applying the binomial recurrence and rearranging the terms:

$$P_{FA}(N + 1) - P_{FA}(N) = \sum_{k=0}^{N} q_N(k)[a_{k,N+1} - a_{k,N}] - p_g \sum_{k=0}^{N} a_{k,N+1}[q_N(k) - q_N(k - 1)].$$

We analyze the terms in this expression to determine the sign of the difference. The first term, $\sum q_N(k)[a_{k,N+1} - a_{k,N}]$, is also non-negative due to the monotonicity of $a_{k,N}$ as established in Eq. equation 4. The sign of the final term depends on the difference $q_N(k) - q_N(k - 1)$, which is positive when $k < (N + 1)p_g$ and negative when $k > (N + 1)p_g$.

For a hard prompt, $p_g$ is very small, causing the mode of the binomial distribution, $(N + 1)p_g$, to be close to zero. This means that for the vast majority of the sum (i.e., for $k \geq 1$), the term $q_N(k) - q_N(k - 1)$ is negative, making the third term, $-p_g \sum a_{k,N+1}[q_N(k) - q_N(k - 1)]$, predominantly positive. Even in the case where $k < (N + 1)p_g$ and the term is negative, its magnitude is scaled by $p_g$, which is close to zero. Consequently, its effect is dominated by the first two non-negative terms. Since the significant terms in the expression are non-negative and the negative contributions are negligible, the difference $P_{FA}(N + 1) - P_{FA}(N)$ is positive for small $p_g$. This confirms our claim that for hard prompts, the probability of a false acceptance increases with $N$. We will validate this finding empirically in Section 6.2.1.

## 5.2 The Best of mini-N in-loop Method

Having established the risks of standard BoN sampling, we now introduce our solution: best of mini-N in-loop. The fundamental idea is to partition a large generation budget, $N$, into $L$ sequential loops, each

generating a smaller batch of $n$ candidates where $N = n \times L$. After each loop, the single best response found so far is checked against a threshold to determine if the process can terminate early.

The power of this framework lies in the choice of this threshold, which allows the method to be configured for two distinct applications.

To create a robust alignment guardrail, we use a pre-calculated sequence of progressively adjusted thresholds, $\{\tau_{N_l}\}_{l=1}^{L}$. This adaptive threshold is designed to maintain a consistent level of reliability as the total sample size grows, directly controlling the risk of a false acceptance. The precise methodology for calibrating these thresholds is detailed in Section 5.3.

Conversely, to configure the method as a fast inference accelerator, we use a simple fixed threshold of $\tau = 0$. This setting is designed to maximize the chance of an early exit as soon as any response is deemed acceptable, thereby minimizing average inference time for speed-critical applications.

If all loops complete without a candidate meeting the specified threshold, the system can either return the best response found or abstain entirely—a crucial feature for the guardrail configuration. The precise mechanics of this strategy are detailed in Algorithm 1.

### 5.3 Calibrating the Thresholds

The effectiveness of the alignment guardrail hinges on its pre-calculated sequence of thresholds, $\mathcal{T} = \{\tau_{n \cdot l}\}_{l=1}^{L}$. This section details our practical, data-driven procedure for setting these values, which is designed to be robust on hard prompts where the risk of false acceptance is highest. The approach involves two key steps. First, we use hypothesis testing to formally identify a set of hard prompts from a larger pool. Second, we construct an empirical reward distribution from the responses generated for these prompts and use it to derive a statistically consistent acceptance threshold for any given sample size, $N$.

#### 5.3.1 Identifying Hard Prompts via Hypothesis Testing

A hard prompt is defined to be one for which the generator policy's probability of producing an acceptable response, $p_g$, is very low. To identify such prompts from a large pool of candidates, we frame the problem as a statistical hypothesis test. We first set a minimum acceptable goodness probability, $p_{min}$, that we believe any reasonable policy should achieve (e.g., $p_{min} = 0.05$). For any given prompt, we want to test if its true $p_g$ is below this threshold.

The procedure is as follows:

1. **Set Hyperparameters:** Choose a minimum acceptable response probability, $p_{min}$, and a statistical significance level, $\alpha$ (e.g., $\alpha = 0.01$).

2. **Collect Samples:** For each candidate prompt $x$, generate a large number of i.i.d. responses, $B$, from the policy $\pi(\cdot|x)$.

3. **Count Successes:** Using our trained reward model, count the number of acceptable responses, $T(x) = \sum_{i=1}^{B} \mathbb{I}(\hat{R}_\theta(x, y_i) > 0)$.

4. **Perform a Binomial Test:** We test the null hypothesis that the prompt's true goodness probability is higher than our minimum acceptable level ($H_0 : p_g(x) > p_{min}$). We calculate the p-value of observing a count as low as $T(x)$:

$$\text{p-value} = \Pr(\text{Bin}(B, p_{min}) \leq T(x)) \tag{5}$$

   If this p-value is less than our significance level $\alpha$, we reject the null hypothesis and classify the prompt as hard, adding it to our set $\mathcal{D}_h$.

This process provides a collection of rewards generated specifically from prompts where the policy is known to struggle.

### 5.3.2 Calculating the Threshold from the Empirical CDF

With the rewards from the hard prompts collected, we construct an empirical Cumulative Distribution Function (CDF), denoted $\tilde{F}(r)$, which represents the proportion of collected reward scores less than or equal to a value $r$. This empirical distribution is the foundation for our threshold calibration. The key to our approach is the value $\tilde{F}(0)$. In our framework, a score of zero is the dividing line between an acceptable and unacceptable response. Therefore, $\tilde{F}(0)$ represents the observed probability that a single response drawn from a hard prompt is deemed unacceptable. We set this as our baseline reliability level.

Our goal is to find an adjusted threshold, $\tau_N$, that maintains a consistent, pre-defined level of reliability as the sample size $N$ grows. To ensure the resulting guardrail is robust, our calibration strategy is intentionally conservative. We derive the thresholds from an empirical reward distribution constructed exclusively from hard prompts—those for which the generator is least likely to produce an acceptable response ($p_g \approx 0$). This focus on the worst-case distribution is a direct response to our earlier finding that the reliability of standard BoN degrades most significantly on such prompts. From order statistics, we know that the CDF of the maximum reward from $N$ samples is given by $[\tilde{F}(r)]^N$. We want to find the threshold $\tau_N$ where the probability of the BoN winner being unacceptable is equal to our baseline reliability level, $\tilde{F}(0)$. We achieve this by solving the equivalence:

$$[\tilde{F}(\tau_N)]^N = \tilde{F}(0) \tag{6}$$

Solving for $\tau_N$ yields our final formula for the adjusted threshold:

$$\tau_N = \max\left\{\tilde{F}^{-1}\left([\tilde{F}(0)]^{1/N}\right), 0\right\} \tag{7}$$

Here, $\tilde{F}^{-1}(q)$ is the empirical quantile function, which finds the reward value at the $q$-th percentile of our collected data. The addition of the $\max\{\cdot, 0\}$ operation ensures the calculated threshold $\tau_N$ is always non-negative. This is a critical safeguard. Since a score of zero is the dividing line for acceptability, a negative threshold would defeat the purpose of the alignment guardrail by permitting the acceptance of responses that the model itself scores as unacceptable. This data-driven procedure therefore provides robustly calibrated and conceptually sound guardrail thresholds for use in Algorithm 1. Crucially, because the calibration is performed on the empirical distribution of the normalized reward $R$, all thresholds $\tau_N$ operate on this same scale and are not tied to any separate estimate of $C(x)$.

### 5.4 Two Configurations: The Alignment Guardrail and the Inference Accelerator

The best of mini-N in-loop method offers a clear advantage over standard BoN by allowing practitioners to configure its behavior for their application's needs. The choice of threshold creates a direct trade-off, tuning the framework to act as either a robust alignment guardrail or a fast inference accelerator. It is important to note that both configurations may result in a slightly lower maximum reward compared to a full Best-of-N, as the early-exit condition prioritizes finding an acceptable, rather than the absolute best, response.

**The Alignment Guardrail ($\tau_N$):** This configuration uses the pre-calculated, adaptive threshold, $\tau_N$, detailed in Section 5.3.

- **Primary Goal:** To maximize **reliability** by minimizing the risk of false acceptances.

- **Benefit:** Provides a robust guardrail with a low and stable False Positive Rate, ensuring that approved responses are highly trustworthy.

- **Cost:** May be slightly slower on average than the accelerator configuration, as the higher threshold requires a better response to trigger an early exit.

- **Use Case:** Ideal for applications where correctness and **reliability** are critical. For example, in customer-facing AI, medical information bots, or systems handling sensitive tasks, it is more important to avoid providing a misaligned or harmful answer than to respond as quickly as possible.

**The Inference Accelerator ($\tau = 0$):** This configuration uses a fixed threshold of zero.

- **Primary Goal:** To maximize **speed** by finding any acceptable answer as quickly as possible.

- **Benefit:** Achieves the lowest average inference time due to the high probability of an early exit.

- **Cost:** Sacrifices the reliability guarantees of the guardrail. The risk of false acceptances is significantly higher, making it unsuitable for applications requiring high trustworthiness.

- **Use Case:** Suitable for applications where speed is paramount and the cost of a suboptimal or slightly misaligned response is low. For example, generating creative text, summarizing non-critical documents, or formatting outputs where the factual content is generally correct but the user's stylistic preference might not be perfectly met.

This flexibility allows practitioners to tune the algorithm's behavior, establishing a clear trade-off between the robust guarantees of the alignment guardrail and the raw speed of the inference accelerator.

# 6 Experiment

This section presents the experimental validation for our framework. We begin by detailing the setup, including the models, datasets, and the multi-stage procedure for training our reward model. We then present our results, which are structured to first empirically validate the central problem motivating this work, and then to demonstrate the effectiveness of our solution. First, we provide evidence confirming that the reliability of standard BoN sampling degrades for hard prompts as $N$ increases. Having established this failure mode, we then evaluate the two configurations of our best of mini-N in-loop method: we test the alignment guardrail ability to control false acceptances, and the inference accelerator capacity to reduce inference latency.

## 6.1 Experimental Setup

Our experimental design is inspired by the methodology of Chowdhury et al. (2024). All models were trained using Lora (Hu et al., 2022) on a single NVIDIA T4 GPU.

### 6.1.1 Models and Datasets

- **Base Language Model:** We use `google/gemma-3-270M` (Team, 2025) as the base model for all fine-tuning tasks.

- **Ground Truth Reward Model:** To simulate human preferences and generate a synthetic training dataset, we use `cardiffnlp/twitter-roberta-base-sentiment-latest`, a powerful sentiment analysis model (Camacho-Collados et al., 2022; Loureiro et al., 2022).

- **Dataset:** All experiments use the `stanfordnlp/imdb` dataset (Maas et al., 2011), which contains movie reviews.

### 6.1.2 Training Procedure

The creation of our reward model involves a three-stage process:

1. **Supervised Fine-Tuning (SFT):** We first fine-tune the base Gemma model to generate positive-sentiment text. We use the final 12,000 positive reviews from the IMDB training set, splitting them into 10,000 for training and 2,000 for evaluation. For each review, the first 21 tokens (including the 'bos' token) are used as the prompt, and the model is trained to generate the remainder with a maximum length of 196 tokens.

2. **Synthetic Preference Data Generation:** We use the first 12,000 negative reviews from the IMDB dataset as prompts for our SFT model. For each prompt, we generate two distinct responses $(y_1, y_2)$, with a maximum length of 196 tokens.

   We then score these responses using the ground-truth RoBERTa model. We extract the probability of the 'positive' label and rescale it to a range of $[-1, 1]$ by applying the transformation $f(p) = 2(p-0.5)$. A positive score indicates an acceptable response, while a negative score indicates an unacceptable one.

   Finally, we simulate a human labeller's choice by adding i.i.d. Gumbel noise to the ground-truth scores of $y_1$, $y_2$, and the outside option (with a score of 0). The option with the highest noisy score is recorded as the choice, $d \in \{0, 1, 2\}$ as illustrated by Eq. equation 1. This process yields a dataset of tuples: $(x, y_1, y_2, d)$.

3. **Reward Model Training:** The final reward model is trained on 10,000 samples from this synthetic preference dataset, with 2,000 samples held out for evaluation. The model is trained to maximize the log-likelihood as defined in Eq. equation 3.

### 6.1.3 Guardrail Threshold Calibration

To calculate the adjusted thresholds, $\tau_N$, for our guardrail experiments, we followed the practical approximation method detailed in Section 5.3. We used the first 500 prompts from the IMDB test set as our candidate pool for identifying hard prompts.

For each candidate prompt, we performed a binomial hypothesis test by generating $B = 150$ responses from our fine-tuned policy. We set a minimum acceptable goodness probability of $p_{min} = 0.05$ at a significance level of $\alpha = 0.01$. Prompts that were statistically likely to have a true goodness probability lower than $p_{min}$ were classified as hard.

The reward scores from all responses generated for these identified hard prompts were collected to construct an empirical CDF. This distribution was then used to calculate the $\tau_N$ values required for our experiments. The resulting thresholds are shown in Table 1.

Table 1: Calculated guardrail thresholds ($\tau_N$) for different sample sizes ($N$). These values are used in the Best of mini-N in-loop experiments.

| Sample Size (N) | 4 | 8 | 12 | 16 | 20 | 24 | 28 | 32 |
|---|---|---|---|---|---|---|---|---|
| Threshold ($\tau_N$) | 0.260 | 0.380 | 0.517 | 0.599 | 0.659 | 0.698 | 0.727 | 0.748 |

## 6.2 Results

Our experimental results validate the two primary claims of this paper. First, we demonstrate that standard BoN sampling suffers from a critical reliability issue, with the rate of false acceptances increasing alongside the sample size. Second, we show that our best of mini-N in-loop strategy provides a robust solution by demonstrating the effectiveness of its two configurations: the alignment guardrail and the inference accelerator. All experiments were conducted on a held-out slice of 1,000 prompts from the IMDB test set.

### 6.2.1 Standard Best-of-N is Unreliable for Hard Prompts

Our first experiment confirms the central problem motivating our work: standard BoN is prone to an increasing number of false acceptances as the sample size, $N$, grows. Figure 1 illustrates this trade-off. While the mean ground-truth reward improves with larger $N$ (the green line), this perceived increase in quality comes at a steep cost. The absolute count of false positives—unacceptable responses incorrectly rated as good—inflates significantly, rising from 104 at N=1 to 210 at N=32 (the red line). This demonstrates that for hard prompts, the standard BoN approach becomes progressively less reliable, amplifying the risk of showing a low-quality or inappropriate response to the user.

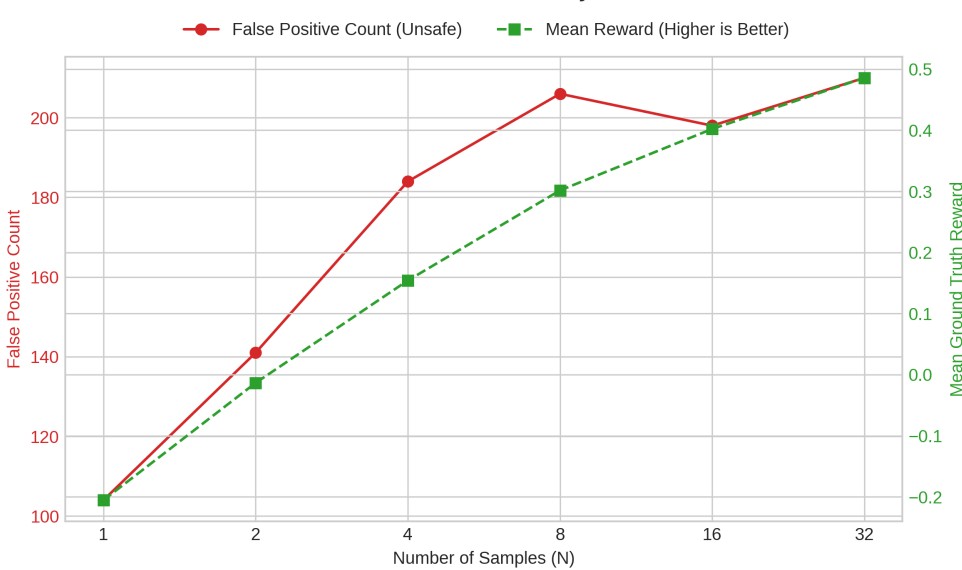

Figure 1: In standard BoN, the False Positive Count increases with the number of samples ($N$), even as the mean reward improves. This highlights a critical reliability vulnerability.

## 6.3 Best of mini-N in-loop: Validating the Two Configurations

Our second set of experiments demonstrates that our best of mini-N in-loop strategy offers a superior alternative to standard BoN sampling. By analyzing the performance of its two distinct configurations—the alignment guardrail and the inference accelerator—we show that our framework allows practitioners to successfully optimize for either reliability or speed, outperforming the monolithic baseline in both cases.

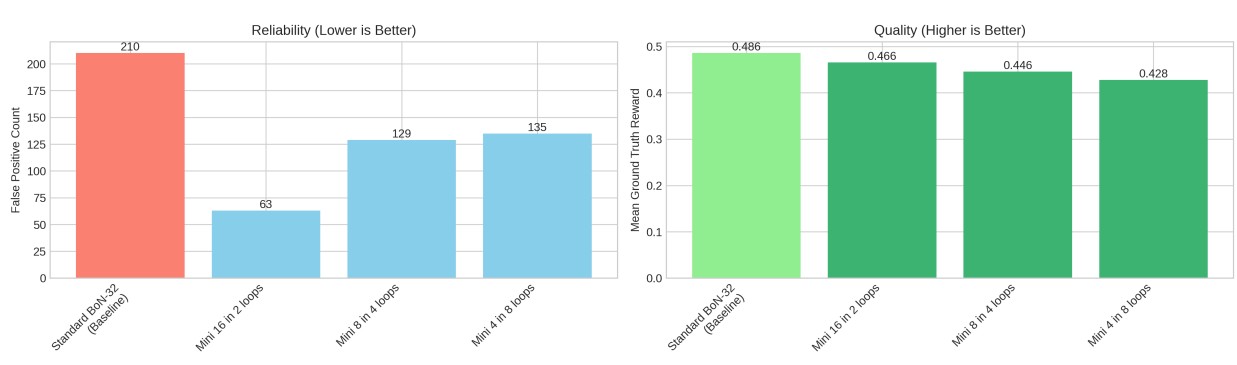

Figure 2: Performance of the alignment guardrail configuration compared to the BoN-32 baseline. The Mini-16 in 2 loops setting dramatically reduces the False Positive Count with only a marginal decrease in mean reward.

**The Alignment Guardrail: A 70% Reduction in Reliability Failures**  The primary advantage of our method is its ability to serve as a robust guardrail against unacceptable responses. As shown in Figure 2, the alignment guardrail configuration is highly effective. The Mini-16 (2 loops) setting reduces the number of false positives by a remarkable **70%**, from 210 in the baseline to just 63. Critically, this substantial

reliability gain is achieved with a negligible trade-off in response quality, with the mean ground truth reward dropping only marginally from 0.486 to 0.466.

The detailed metrics in Table 2 reveal how this is achieved. The guardrail configuration significantly boosts **Precision** to **94.3%** (from 88.0%) and slashes the **False Positive Rate (FPR)** to just **15.8%** (from 54.1%). This confirms the guardrail's effectiveness at correctly identifying and approving high-quality responses while rejecting those that fail to meet the acceptability threshold.

Inference Accelerator Performance Comparison at N=32 Budget

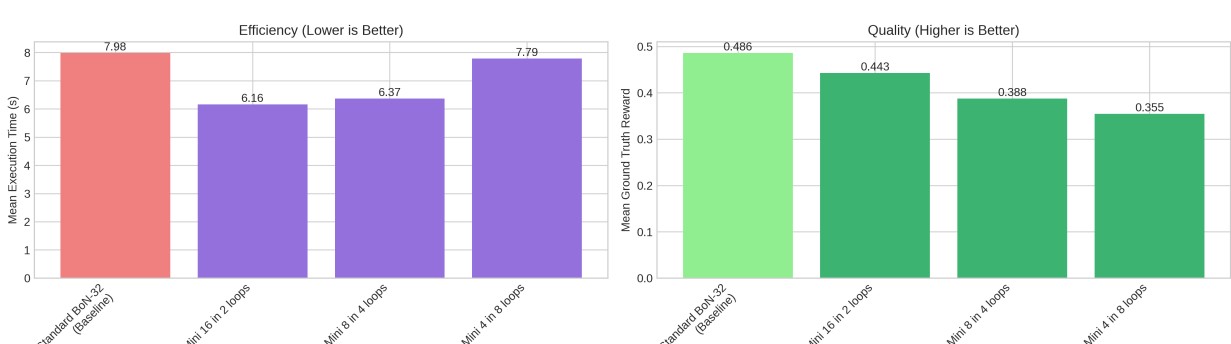

Figure 3: Performance of the inference accelerator configuration. The Mini-16 in 2 loops setting provides the fastest mean execution time, outperforming the BoN-32 baseline by over 22%.

**The Inference Accelerator: Over 22% Faster Inference**   In scenarios where speed is the priority, the same framework can be configured as a powerful inference accelerator. Figure 3 shows that our method again outperforms the standard BoN baseline. The Mini-16 (2 loops) setting is the fastest of all configurations, reducing the average execution time by over **22%**—from 7.98 seconds for the baseline to 6.16 seconds.

As shown in Table 2, this speed advantage is achieved by maintaining a high **Recall** of **95.4%**, equivalent to the baseline. This indicates that the accelerator is just as effective at finding an acceptable response, but its early-exit mechanism allows it to do so much more efficiently. This establishes our method as a clear and tunable framework for achieving significant latency improvements with only a minimal trade-off in the final response quality.

Table 2: Detailed performance metrics for key methods at a total budget of $N = 32$. The table highlights the trade-offs between our two configurations: the alignment guardrail enhances reliability by maximizing Precision and minimizing the FPR, while the inference accelerator achieves its speed by maintaining a high Recall.

| Method | TP | FP | TN | FN | Precision | Recall | FPR |
|---|---|---|---|---|---|---|---|
| *Baseline* | | | | | | | |
| Standard BoN-32 | **1539** | 210 | 177 | 74 | **88.0%** | **95.4%** | 54.1% |
| *Alignment Guardrail* | | | | | | | |
| Best of mini-16 (2 loops) | 1053 | **63** | **336** | 548 | **94.3%** | 65.7% | **15.8%** |
| *Inference Accelerator* | | | | | | | |
| Best of mini-16 (2 loops) | 1510 | 225 | 192 | **73** | 87.0% | **95.4%** | 54.0% |

# 7   Conclusion

In this work, we addressed a fundamental limitation of preference alignment techniques that rely on pairwise comparisons: their inability to distinguish between what is merely *better* and what is genuinely *good enough.*

We demonstrated that this gap leads to critical reliability failures in Best-of-N (BoN) sampling, where systems may select the least bad of many unacceptable options. Our primary contribution is a new reward modeling framework that, by incorporating an outside option into a choice-theoretic model, shifts the paradigm from learning relative rankings to capturing a direct signal of contextual quality.

This new reward model enables our main practical contribution: Best of mini-N in-loop, a single, adaptive inference framework that can be configured for two distinct applications. Our experiments validated both configurations, showing that when tuned as an alignment guardrail, it reduces reliability failures by a remarkable 70%. When tuned as an inference accelerator, it improves average inference speed by over 22% compared to the standard BoN baseline.

Our framework provides practitioners with a principled and tunable method for navigating the critical trade-off between reliability and computational efficiency. Future work could extend this approach to more complex, multi-turn conversational settings or reasoning tasks. Furthermore, exploring the synergies between our method and other inference-time techniques, such as speculative decoding, represents a promising avenue for developing even more reliable and efficient large language models.

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

# A   Appendix: Best of Mini N in Loop Algorithm

---

**Algorithm 1** Best of mini-N in-loop

---

1: **Input:** Prompt $x$, total budget $N$, mini-batch size $n$, number of loops $L = N/n$, tuning mode $m \in$ {'Alignment Guardrail', 'Inference Accelerator'}.
2: **Initialize:** Set of all responses $\mathcal{Y}_{all} \leftarrow \emptyset$.
3: **Initialize:** Maximum observed reward $\hat{R}_{max} \leftarrow -\infty$.
4: **Initialize:** Best response found $y^* \leftarrow$ None.
5: **Initialize:** Loop success flag *found_acceptable* $\leftarrow$ False.
6: **if** $m =$ 'Alignment Guardrail' **then**
7:     **Pre-calculate thresholds:** $\mathcal{T} = \{\tau_{n \cdot l}\}_{l=1}^{L}$ using the method in Section 5.3.
8: **else**
9:     **Set constant threshold:** $\mathcal{T} = \{0\}_{l=1}^{L}$.
10: **end if**
11: **for** $l = 1$ **to** $L$ **do**
12:     Let current total sample size be $N_l = n \times l$.
13:     Retrieve the pre-calculated threshold for this stage, $\tau_{N_l}$, from $\mathcal{T}$.
14:     Generate a new mini-batch of $n$ responses: $\mathcal{Y}_{new} = \{y_1, \ldots, y_n\} \sim \pi(\cdot|x)$.
15:     Update the set of all responses: $\mathcal{Y}_{all} \leftarrow \mathcal{Y}_{all} \cup \mathcal{Y}_{new}$.
16:     Find the best response so far: $y^*_{current} = \arg\max_{y \in \mathcal{Y}_{all}} \hat{R}_\theta(x, y)$.
17:     Update the maximum reward: $\hat{R}_{max} = \hat{R}_\theta(x, y^*_{current})$.
18:     Update the best response: $y^* \leftarrow y^*_{current}$.
19:     **if** $\hat{R}_{max} > \tau_{N_l}$ **then**
20:         *found_acceptable* $\leftarrow$ True.
21:         **break**                                   ▷ Early exit: an acceptable candidate has been found.
22:     **end if**
23: **end for**
24: **if** *found_acceptable* **then**
25:     **Return:** $y^*$                              ▷ Return the best acceptable response.
26: **else**
27:     **Return:** $y^*$ or None       ▷ Optionally, refuse to respond if no candidate met the final threshold.
28: **end if**

---

