# OpenReview forum: "Best of mini-N in-loop Sampling: A Contextual Quality Reward Model for Reliable and Efficient Best-of-N Sampling"
_TMLR — Rejected by TMLR_

### Review · Reviewer_ycha · 2025-11-29

**Summary Of Contributions:**

This article proposes a framework of data collection and modeling for controlling the quality of question answer acceptability by learning a normalized reward model. A best-of mini-N sampling scheme is then proposed to reduce false acceptable answers when acceptable answers are hard to draw from a policy. Compared to exiting best-of N sampling scheme, this method can reduce reliability failure or improve inference speed to draw acceptable answers. These ideas are evaluated on an synthetic experiment to show the effectiveness of the proposed framework.

**Additional Comments:**

I do not have enough time to read Section 5.1 and 5.3, 6.2 in detail.

**Audience:**

Yes

**Audience Explanation:**

The topic is closely related to the quality control of ChatGPT.

**Claims And Evidence:**

No

**Claims Explanation:**

The article is mostly well written. However, some key elements about the framework formulation, as well the experimental setting is missing. These make the article not so easy to follow.

**Requested Changes:**

* A central idea of the framework is to consider an outside option as an explicit label when one aim to learn a normalized reward model. The reward is normalized so that it equals to zero on the outside option (below eq 2). However, it is not clear whether the learnt reward model R_theta(x,0) = 0 for any x and theta. A clarification is needed as it seems that this condition is a key to learn to reject unacceptable answers based on eq 3.
* Details about how R_theta is constructed is missing in the numerical experiment section (6.1.2). This should be detailed.
It is proposed in Section 5.3.1 a hypothesis testing scheme. This scheme relies on an estimated reward R_theta which can be inaccurate. It is not clear how this inaccuracy could affect the power of the hypothesis testing. Is there any statistical analysis on this?
* More details could be included in Section 6.1.1 to describe the dataset and setup. The Ground Truth Reward Model is a binary classifier to decide whether y acceptable given x? A review is acceptable means that is a positive review? This seems to be inconsistent with the standard setting that reward models are trained with pairwise comparison data.
* How do you design the data collection scheme to build a training dataset in practice? The choice data in the numerical experiment section is built artificially from a synthetic dataset. It is not so clear how this could be done as it seems to depend on how many answers one want to draw from a policy pi(.|x).
* Some notations should be adjusted to make them consistent in the article. For example, the use of d_i in eq. 2 is not the same as the one in Section 4.1. This is very confusing to read. Some text contains Eq. equation *, you could the Eq. as it is redundant.

---

### Review · Reviewer_qgsD · 2025-12-27

**Summary Of Contributions:**

* Identifies a reliability failure in standard Best-of-N (BoN) sampling: as $N$ increases, the probability of selecting an unacceptable response ("false acceptance") increases for hard prompts.
* Proposes a "Choice-Based Reward Model" trained with an explicit "outside option" (reject all) to learn an absolute acceptability threshold.
* Introduces "Best of mini-N in-loop," an inference strategy that partitions generation into sequential loops and uses the acceptability signal to trigger early exits.
* Validates the method with a 270M parameter model on the IMDB sentiment task. Reports
  * improved reliability / reduced false positives (“Guardrail” mode)
  * inference speedups (“Accelerator” mode)

**Additional Comments:**

None.

**Audience:**

Yes

**Audience Explanation:**

The theoretical contribution—incorporating an "outside option" into the reward model to capture absolute acceptability—is novel, elegant, and theoretically sound (based on Discrete Choice Theory). This addresses the real and practical problem of "least bad" selection in BoN sampling. If the experimental validation were robust, this would be a valuable contribution to the alignment literature.

**Broader Impact Concerns:**

None.

**Claims And Evidence:**

No

**Claims Explanation:**

1. **Baseline Validity & Efficiency Claims:**
   1. The paper claims a 22% improvement in inference speed using sequential loops (e.g., 2 loops of 16\) compared to a baseline of $N=32$. However, standard Best-of-N is typically implemented with **batch parallelism**, generating all $N$ candidates simultaneously in a single pass to maximize GPU throughput.
   2. For the small model used (270M parameters) on a T4 GPU, a batch size of $N=32$ fits easily in memory. In a fully parallelized setting, the wall-clock latency is determined by the autoregressive decoding length, not linearly by the batch size (until compute saturation). Therefore, splitting the batch into two sequential loops ($16 \\to \\text{Check} \\to 16$) introduces a synchronization barrier that should theoretically ***increase*** latency compared to a single parallel batch of 32\.
   3. The reported speedup suggests the baseline may have been implemented serially or inefficiently, or that "speed" refers to total compute rather than user-facing latency. Without clarification, the efficiency claims are not convincing.
2. **Insufficient Experimental Scale:**
   1. The validation relies exclusively on a `gemma-2-270M` model and a simple IMDB sentiment task where the reward signal is synthetic (simulated by a RoBERTa classifier). This "toy" setting is insufficient to support general claims about LLM alignment reliability. Reward modeling dynamics, "false acceptance" rates, and computational bottlenecks differ significantly in realistic settings (e.g., 7B+ models, complex instruction following, or reasoning tasks).
3. **Missing Reproducibility Details:**
   1. The submission lacks an appendix with critical details regarding the inference infrastructure. Specifically, the method of handling batching for the baseline and the "mini-N" method is not described, making it impossible to verify the latency claims.

**Requested Changes:**

**Critical adjustments:**

1. **Efficiency**
   1. Explicitly clarify how the baseline $N=32$ was generated. Was it a single parallel batch or a serial sequence?
   2. If it was parallel, please explain how a sequential loop method achieves lower latency (wall-clock time) given that modern GPUs can likely decode 32 sequences in roughly the same time as 16 for a 270M model.
   3. If the baseline was serial, please justify why this is a valid comparison, or re-run the baseline using standard batch parallelism.
2. **Scale**
   1. Validate the method on a model of at least standard research size (e.g., \~7B parameters) and a more complex task (e.g., AlpacaEval, GSM8K, or IFEval).
   2. The current evidence on a 270M sentiment model with synthetic rewards is insufficient to support the broad claims of "reliable and efficient" alignment.
3. **Details/Appendix**
   1. Provide a complete appendix with training hyperparameters, prompt templates, and the exact hardware/software configuration (including batching implementation) used for the latency benchmarks.

**Strengthening suggestions:**

4. Expand the analysis of the "mini-N" settings to better characterize the trade-off surface between latency and reliability (currently only 3 data points are shown).

---

### Review · Reviewer_PNVJ · 2026-01-02

**Summary Of Contributions:**

**Contributions**:

This paper identifies a reliability issue in standard Best-of-N (BoN) selection with reward models trained solely on pairwise preferences: on difficult prompts, it can select the “least bad” yet still unacceptable candidate, leading to elevated false acceptance rates.

To address this, it augments preference data with an explicit outside option (“None of the above is acceptable”) and trains a discrete choice model anchored at this option, yielding an acceptability scale where R(x,y)>0 indicates the candidate is preferred over abstention. Building on this reward, the paper introduces a best-of-mini-N in-loop sampling procedure that allocates a fixed sampling budget across multiple mini-batches, configurable as (i) an alignment guardrail that calibrates thresholds to control false positives, or (ii) an inference accelerator that reduces average sampling cost via early stopping.

Finally, it proposes an empirical threshold calibration scheme and validates the approach with Gemma-3-270M in a controlled sentiment-generation setting, showing that the guardrail mode reduces false positives while the accelerator mode lowers average computation.

**Strengths**:

-The outside option provides a natural “zero point” of acceptability, making threshold-based rejection/abstention easy to interpret and implement.

-The method supports (i) a guardrail mode to reduce false positives and (ii) an accelerator mode to reduce average compute via early stopping.

**Weaknesses**:

-The evaluation is largely confined to a single sentiment-generation task with synthetic preference construction. It is therefore unclear how well the approach transfers to more diverse instruction-following tasks (e.g., factuality, safety, multi-constraint prompts).

**Audience:**

Yes

**Audience Explanation:**

The paper highlights an important failure mode of best-of-N under pairwise-preference reward models—false acceptances can increase with N on hard prompts, which is relevant to decoding-time alignment and guardrail design.

**Broader Impact Concerns:**

None. I do not have broader impact concerns for this submission.

**Claims And Evidence:**

Yes

**Claims Explanation:**

Experiments show substantial reductions in false positives under the guardrail configuration and notable reductions in average sampling cost under the accelerator configuration.

**Requested Changes:**

If feasible, adding experiments on one or two additional tasks would further strengthen the empirical support and generality of the conclusions.

---

### Review · Reviewer_mLNo · 2026-01-05

**Summary Of Contributions:**

This paper identifies a critical limitation in Best-of-N (BoN) sampling: reward models trained on pairwise preferences lack an absolute measure of quality, often leading to the selection of the "least bad" option on hard prompts. To address this, the authors propose a discrete-choice reward modeling framework that incorporates an "outside option" (reject all) during training, enabling the model to learn a contextual acceptability threshold. Leveraging this signal, they introduce Best of mini-N in-loop, an adaptive inference strategy that processes candidates in sequential batches. This method features an early-exit mechanism that can be configured either as an "Alignment Guardrail" (to minimize false acceptances) or an "Inference Accelerator" (to reduce latency).

**Strengths**

Adding an outside option is a clean way to inject the missing “none of these are acceptable” signal into reward modeling, and it makes the accept/reject logic at inference time feel much more principled than ad-hoc score thresholds.

The inference-time procedure is also simple to implement.

The paper is fairly readable overall, and although I believe shortening sections before Section 5 could be helpful to make central ideas more pointed.


**Weaknesses**

The biggest issue for me is that the experimental setup is very synthetic and very matched to the modeling assumptions. In the data generation, the “labeler” is simulated by adding i.i.d. Gumbel noise and then choosing the max, which makes the results less convincing as evidence that the approach will hold up with real annotators.

Second, the “inference accelerator” speed claim needs a lot more context. In practice, standard BoN is usually done with batch parallelism, and for small models (like 270M) you often don’t pay 2× latency going from batch 16 to batch 32, for example. So a sequential 2-loop approach *can* be faster for early exit, but the paper needs to be very explicit about how the baseline was run and what “speed” actually measures (wall-clock latency vs total compute).

Third, baseline coverage feels a bit thin. If the key contribution is “outside option makes acceptability meaningful,” I really want to see a strong baseline where you take a standard pairwise reward model and do the most reasonable calibration you can for accept/reject. Right now the paper mostly compares to a plain BoN baseline, which makes it hard to know how much of the win is the outside option vs just “better calibration / better thresholding.” The baseline could just be:

**Audience:**

Yes

**Audience Explanation:**

People working on reward modeling, answer refusal, and test-time compute scaling might care about this.

**Claims And Evidence:**

No

**Claims Explanation:**

In the authors’ controlled IMDB sentiment setup, the numbers look encouraging. But because the whole pipeline is built in a very friendly synthetic regime, I don’t think the current experiments support the broader “reliable and efficient” story for real alignment workloads.

**Requested Changes:**

I think the paper would be much stronger with the following additions. I’m intentionally asking for things that can be run and interpreted cleanly.
**1) One realistic evaluation using an existing outside-option dataset (HelpSteer2-Preference).**
The paper already cites HelpSteer2-Preference as having a “Neither response is valid” label. A concrete experiment would be:
- Train the proposed choice-based reward model using that label as the outside option (do not drop those samples).
- Evaluate on a held-out set of prompts from the same source/domain. For each prompt, generate N candidates (e.g., N=32) from a base LM, and compare BoN vs mini-N-in-loop (guardrail + accelerator).
- For evaluation, it’s enough to label only the *final selected outputs* from each method (not all N candidates). Even a small human study (e.g., a few hundred prompts, 2–3 annotators, majority vote accept/reject) would directly test whether the method reduces “bad outputs that still get selected.”

**2) Misspecification / robustness study (in your existing synthetic setup).**
Keep the current IMDB pipeline, but change the simulated labeling noise away from i.i.d. Gumbel/logit to other noises, e.g., heavier-tailed noise. Report how much the guardrail and accelerator benefits degrade. This is helpful in addressing whether the method is fragile to the feedback model being wrong.

**3) Ablate the outside option with a clearly-defined calibrated pairwise baseline (can be an example I drafted above in the Weaknesses part).**

**4) Elaborate the efficiency evaluation.**
Give details about, for example, hardware/software, batching strategy, and whether “speed” is wall-clock latency or total compute. Also rerun the BoN baseline in a standard parallel-batch way (if it wasn’t already), so the comparison is fair.

**5) One scaling check on a larger LM.**
Even a single run on a 7B-ish model (or the closest feasible size) would help, because throughput/latency behavior and reward model behavior can look different from a 270M setup.

---

### Comment · Action_Editor_erVG · 2026-01-05
**Author Response**

There are now sufficient reviews for the authors to gain an understanding of the key concerns regarding the initial submission. There are concerns regarding experiments being sufficient to justify the claims, reproducibility, as well as whether they outperform strong enough baselines. These issues should be covered in the response/revision.

---

### Decision · Action_Editor_erVG · 2026-02-24

**Recommendation:** Reject

**Additional Comments:**

This paper proposes a clean and principled idea: incorporate an explicit outside option into reward modeling to give Best-of-N decoding an absolute notion of acceptability. Conceptually, this is elegant. The problem is not the idea—it is the evidentiary standard and whether the claims can be justified with respect to how well-studied this topic is in recent years. In particular, several reviewers independently identified core gaps:

* Synthetic and narrow evaluation (mLNo, qgsD, PNVJ): The empirical validation is confined to a 270M model and a synthetic sentiment setup with simulated noise. While coherent in this controlled regime, it does not establish robustness under realistic alignment settings or modern model scales.

* Questionable efficiency comparison (qgsD): The claimed speedup is benchmarked against a baseline whose batching strategy is unclear. Without a fully parallelized Best-of-N comparison, the inference advantage remains unconvincing.

* Baseline isolation (mLNo): The contribution of the outside option is not cleanly disentangled from improved calibration or thresholding of a standard pairwise reward model.

* Formal clarity and reproducibility (ycha): Key aspects of the normalization constraint and inference procedure remain under-specified.

* More broadly, the high-level mechanism—partitioning rollouts, adaptively allocating sampling budget, and exploiting relative scoring signals—resembles an already visible design pattern in large-scale agent and ensemble decoding practice, including publicly documented multi-policy and rollout-scaling strategies one can find in NVIDIA blogs [  https://blogs.nvidia.com/blog/ai-scaling-laws/  ;  https://developer.nvidia.com/blog/breaking-through-rl-training-limits-with-scaling-rollouts-in-brorl/ ] . The manuscript does not clearly articulate what is fundamentally new beyond this established paradigm.

The idea is promising, but the veracity of the claims in this case is hard to distinguish from their lack of differentiation to prior art, and the concerns of the reviewers. I encourage the authors to revise and resubmit, after: strengthening empirical breadth, clarifying efficiency baselines, and more sharply differentiating the contribution from existing ensemble/multi-policy decoding approaches.

**Audience:**

Yes

**Audience Explanation:**

See additional comments.

**Claims And Evidence:**

No

**Claims Explanation:**

There are limitations.

**Resubmission Of Major Revision:**

The authors may consider submitting a major revision at a later time.